# Learning to Reason for Factuality

**Xilun Chen**[1]  **Ilia Kulikov**[1]  **Vincent-Pierre Berges**[1]  **Barlas Oğuz**[1]  **Rulin Shao**[2]  **Gargi Ghosh**[1]  **Wen-tau Yih**[1]

## Abstract

Reasoning Large Language Models (R-LLMs) have significantly advanced complex reasoning tasks but often struggle with factuality, generating substantially more hallucinations than their non-reasoning counterparts on long-form factuality benchmarks. However, extending online Reinforcement Learning (RL), a key component in recent R-LLM advancements, to the long-form factuality setting poses several unique challenges due to the lack of reliable verification methods. Previous work has utilized automatic factuality evaluation frameworks such as FActScore to curate preference data in the offline RL setting, yet we find that directly leveraging such methods as the reward in online RL leads to reward hacking in multiple ways, such as producing less *detailed* or *relevant* responses. We propose a novel reward function that simultaneously considers the factual precision, response detail level, and answer relevance, and applies online RL to learn high quality factual reasoning. Evaluated on *six* long-form factuality benchmarks, our factual reasoning model achieves an average reduction of **23.1 percentage points** in hallucination rate, a **23%** increase in answer detail level, and no degradation in the overall response helpfulness.[1]

## 1. Introduction

The recent emergence of Reasoning Large Language Models (R-LLMs) such as OpenAI-o1 (OpenAI, 2024) and DeepSeek-R1 (DeepSeek-AI, 2025b), which invoke a Long Chain-of-Thought (Long CoT) thinking process before producing the final response, have significantly advanced LLM's capabilities on complex reasoning tasks such as mathematics and coding. One area that has often been overlooked so far in R-LLM research is *factuality*. This aspect becomes increasingly crucial as the enhanced capabilities of R-LLMs lead to their being entrusted with more complicated and consequential tasks. Unfortunately, it has been suggested that R-LLMs tend to hallucinate *more* than their non-reasoning counterparts (Hughes et al., 2023). We benchmarked two popular R-LLMs, DeepSeek-R1 and QwQ-32B (Qwen-Team, 2025), on six long-form factuality datasets. Our findings indicate that their hallucination rates are, on average, *10 and 13 percentage points higher* than those of DeepSeek-V3 (DeepSeek-AI, 2025a) and Qwen-2.5-32B (Yang et al., 2024), respectively. We hypothesize that this is because existing Reinforcement Learning (RL) training for R-LLMs primarily targets logical reasoning tasks such as math and coding, often overlooking other important properties like factuality. This leads us to the following research question:

> **RQ**: *Can we learn reasoning strategies that improve the factuality of an (R-)LLM?*

Traditionally, RL alignment optimizes for *verifiable rewards* (RLVR, Lambert et al., 2025) in domains such as mathematics and programming, or *human preferences* (RLHF, Ouyang et al., 2022) for general instruction following. In contrast, factuality, especially in **long-form generations**, does not lend itself well to either approach. There is no reliable method to deterministically and accurately verify the factuality of a long-form response, and human verification requires significant manual effort, making it expensive and time-consuming (Min et al., 2023). Although there are automatic evaluation frameworks for long-form factuality, such as FActScore (Min et al., 2023), which have been employed in previous factuality alignment work (Tian et al., 2023; Lin et al., 2024), these methods are limited to offline RL where they create pairwise preference data for Direct Preference Optimization (DPO, Rafailov et al., 2023). In contrast, online RL offers notable advantages: it is integral to recent advances in R-LLMs (DeepSeek-AI, 2025b), and prior work consistently demonstrates the benefits of training on *on-policy* data (e.g. self-generated responses) for improving factuality (Lin et al., 2024; Zhang et al., 2024, *inter alia.*). However, applying online RL to learn *factual reasoning* in long-form responses remains an open problem with several outstanding challenges.

---

[1]FAIR at Meta [2]University of Washington. Correspondence to: Xilun Chen <xilun@meta.com>.

*Proceedings of the 43rd International Conference on Machine Learning*, Seoul, South Korea. PMLR 306, 2026. Copyright 2026 by the author(s).

[1]Our code and data is available at: https://github.com/facebookresearch/factual_reasoning

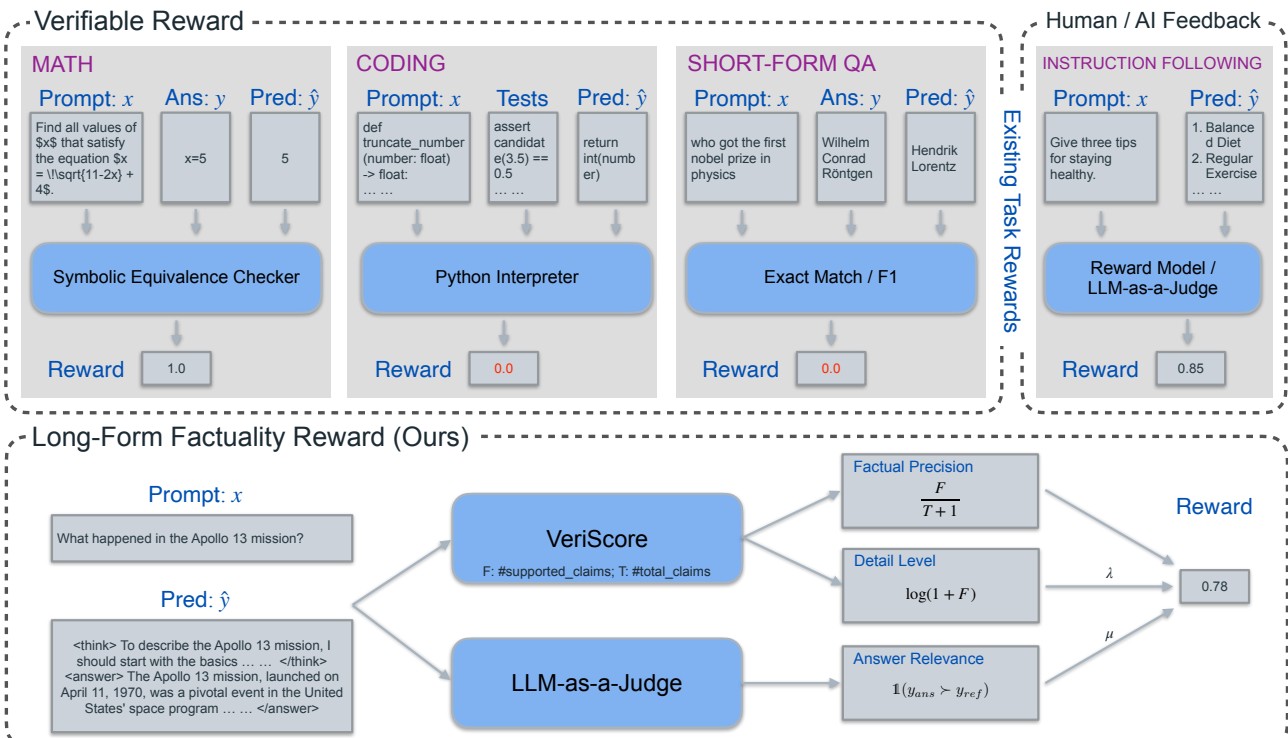

*Figure 1.* Reward design for Long-Form Factuality (bottom). Unlike other tasks (top), the factuality of long-form responses cannot be reliably assessed by rule-based heuristics or an LLM judge. Relying solely on automatic evaluation methods such as VeriScore may lead to less *detailed* or *relevant* responses. We propose a new reward design that simultaneously considers factual precision, response detail level, and answer relevance (Section 3.1).

The first challenge lies in the reward design. In our experiments, we find that optimizing solely towards a factuality reward may result in unintended outcomes. The model learns to produce much shorter and less detailed responses, as a shortcut to achieve higher factual precision, because it is significantly easier for an LLM to generate a single correct fact than to produce a detailed answer containing, for example, 50 facts without any hallucinations, even though both would have a perfect factual precision. Furthermore, even if the reward manages to consider both factuality and the level of detail in the answer, it remains possible to falsely inflate (hack) the reward by producing *less pertinent*, or in extreme cases, *irrelevant* answers. Consider the following extreme example: a model recites the same Wikipedia article, which is both factual and detailed, in response to *every* question it is asked. Such a model would be utterly useless, yet it would achieve very high scores in both factuality and detail level. Last but not least, existing automatic long-form factuality evaluation methods, which typically involve LLM-based atomic claim extraction and verification along with web searches to find relevant evidence documents, are very time-consuming. This makes them unsuitable for real-time reward calculation in online RL. For instance, VeriScore (Song et al., 2024), a recent long-form factuality evaluation method, can take several minutes to verify a single response.

In this work, we propose the first online RL recipe for long-form factuality, with a novel reward function that addresses these challenges. Our factual reasoning reward has three components to mitigate the various ways of hacking the reward described above: it considers (1) factual precision, (2) response detail level, and (3) answer relevance at the same time. For computing (1) and (2) we implement an optimized and scalable version of VeriScore, achieving up to a 30x speedup, which makes it suitable for real-time reward calculation in online RL rollouts. For (3) we combine these rewards with the overall quality of the response measured using LLM-as-a-Judge. We evaluate our method on *six* long-form factuality benchmarks, including Long-Fact (Wei et al., 2024), FAVA (Mishra et al., 2024), AlpacaFact (Dubois et al., 2024), Biography (Min et al., 2023), FactBench (Bayat et al., 2024), and FACTORY (Chen et al., 2025a), showing that our factual reasoning model trained with online RL using GRPO (Shao et al., 2024) achieves an average of **23.1 points** higher factuality precision while producing **23%** more factual statements in the responses, without degradation in the overall response helpfulness (LLM-as-a-judge win rate >50% over the base model).

**Conflict of Interest Disclosure** The authors are employed by Meta, which leads the development of Llama models, which was among the ones evaluated in this paper.

## 2. Offline Training for Factual Reasoning

In the next two sections, we describe approaches to learning factuality-focused Long CoT reasoning for a given *base model*[2]. In this section we first present our approach to curating training data, followed by our recipes for supervised fine-tuning and offline RL via DPO, both of which will serve as baselines. Then, in the next section, we introduce our proposed online RL setup to directly conduct on-policy optimization towards a factuality reward.

### 2.1. Training prompts

For both offline and online RL, a diverse and high-quality set of fact-seeking questions is needed as training prompts. Previous work on factuality alignment either focuses on a specific domain and uses in-domain training prompts (Tian et al., 2023), or relies on filtering from existing datasets such as OpenAssistant (Köpf et al., 2023) as done by (Lin et al., 2024). We attempted a similar approach by prompting an LLM to identify fact-seeking questions from WildChat prompts (Zhao et al., 2024), a large-scale dataset of user-chatbot conversations with a great degree of diversity. However, we observed that it was challenging for an LLM to reliably classify fact-seeking questions from such a diverse set of natural prompts, resulting in noisy outcomes with many low-quality prompts.

Therefore, we adopt a new approach to curating the training prompt set that i) is likely to appear in real-world scenarios, and ii) incentivizes factuality as a major factor of a high-quality response. In particular, we adopt Llama 4[3] to generate synthetic prompts by providing two sets of *grounding prompts* as demonstrations: one set of real-world diverse prompts from WildChat (Zhao et al., 2024) and another set of fact-seeking prompts from the non-test split of LongFact (Wei et al., 2024). The goal is for the model to generate prompts that are diverse and likely asked by real humans, similar to the examples in the first group of grounding prompts, while also requiring factual knowledge to provide a good answer, as seen in the second group of grounding prompts. The full Llama 4 prompt and some examples of the generated questions can be found in Appendix E. We generate a total of 7k synthetic prompts, divided into a 3k SFT split and a 4k RL split.

### 2.2. Supervised Finetuning (SFT)

In preliminary experiments, we find it beneficial to first perform supervised finetuning (SFT) before applying RL algorithms. In particular, in offline DPO experiments, the model

---

[2]In this paper, we use the term *base model* to refer to the model used as initialization during our factual reasoning training, irrespective of whether it has undergone any prior alignment training.

[3]Llama-4-Maverick-17B-128E-Instruct-FP8

struggles to consistently follow the Long CoT reasoning format without SFT, even after additional preference data on format-following was added. On the other hand, SFT effectively teaches the model to follow the Long CoT format to produce a reasoning chain (wrapped in `<think>` and `</think>`) before generating the final answer (wrapped in `<answer>` and `</answer>`). In online RL experiments, while it is possible to directly apply RL to produce Long CoT responses in the correct format, SFT on seed factual reasoning data further provides a useful inductive bias for the later RL stage, which stabilizes training and leads to higher-quality responses in practice.

We create seed SFT data with factuality-focused Long CoT reasoning chains by prompting the base model with manually-written 2-shot examples (full prompt in Appendix F) to generate 10 responses for each training prompt in the SFT split. VeriScore is run on each response and the one with the highest factual precision is chosen as the target for SFT.

### 2.3. Direct Preference Optimization (DPO)

Direct Preference Optimization (DPO, Rafailov et al., 2023) has been employed to improve the factuality in a model's responses (Tian et al., 2023; Lin et al., 2024). We will also consider it as a baseline within our setting. DPO is an offline RL algorithm that learns from preference pairs $y_c \succ y_r$ where the chosen response $y_c$ is considered better than the rejected one $y_r$ for a given prompt $x$, by optimizing the following loss:

$$\mathcal{L}_{\text{DPO}} = -\log \sigma \left( \beta \log \frac{\pi_\theta(y_c|x)}{\pi_{\text{ref}}(y_c|x)} - \beta \log \frac{\pi_\theta(y_r|x)}{\pi_{\text{ref}}(y_r|x)} \right) \tag{1}$$

where $\pi_\theta$ is the current policy model, $\pi_{\text{ref}}$ is a reference model (typically the seed model), $\beta$ is a hyper-parameter controlling the deviation from the reference policy, and $\sigma$ is the logistic function.

To form preference pairs for learning factual reasoning, we can follow a similar approach to the SFT data curation. In particular, we sample 10 responses for each training prompt in the RL split, using the same 2-shot prompt template (Appendix F), and run VeriScore to evaluate the factuality of the responses. Among all possible response pairs, we pick the one with the maximum margin in factual precision between the chosen and the rejected responses, subject to two additional conditions: i) The factual precision margin should be greater than a threshold $\tau_m$; and ii) The lengths of the chosen and rejected responses, $l_c$ and $l_r$, do not differ by too much, satisfying: $\left|1 - \frac{l_c}{l_r}\right| \leq \tau_l$ ($\tau_m = \tau_l = 0.1$ in our experiments). Condition i) ensures that there is sufficient variation in factual precision between the pair so DPO can get meaningful learning signals, whereas ii) requires that the lengths of $y_c$ and $y_r$ are similar, which we find beneficial

in avoiding length hacking (Park et al., 2024). As more factual responses tend to be shorter due to the removal of non-factual statements, the chosen responses are more likely to have a shorter length, leading to DPO exploiting this fact during learning and producing much shorter answers. If no pair meets both conditions for a certain prompt, the prompt is removed from the training set. This results in a total of 3.7k preference pairs for DPO training.

# 3. Online Training for Factual Reasoning

More recently, online RL methods such as GRPO (Shao et al., 2024) have become the *de facto* standard for training reasoning LLMs, and have been shown to significantly outperform offline RL in recent studies (Lanchantin et al., 2025). Online RL has been less explored in the factuality space, especially long-form factuality, due to the challenges associated with automatic evaluation and reward design. In this section, we discuss these challenges and propose a new reward formulation for optimizing long-form factuality in online RL.

## 3.1. Reward Design for Factual Reasoning

Most existing work on learning Reasoning LLMs focuses on *verifiable* tasks, such as mathematics and coding, where the verification of a given answer is both accurate and inexpensive. Outside the reasoning domain, RL alignment on non-verifiable tasks typically leverages a *reward model* that is trained to judge the quality of a response. For factual reasoning, however, where the reward must reflect the factuality of *open-ended long-form responses*, neither approach is suitable. Unlike mathematics, where solutions can be verified through symbolic computation, or code, which can be executed to determine if it produces the desired output, there is no consistent and precise method for verifying the factual accuracy of long-form responses. Additionally, LLM-as-a-judge reward is also inadequate, as even state-of-the-art LLMs struggle to reliably assess the factuality of long-form responses due to their susceptibility to hallucinations.

There are automatic evaluation methods for long-form factuality, such as FactScore (Min et al., 2023), SAFE (Wei et al., 2024), and VeriScore (Song et al., 2024), which mostly follow the same paradigm of breaking down the long-form response into atomic claims and verifying each claim with an LLM against a set of retrieved evidence documents using a retriever or web search engine. Although these methods have become the standard for long-form factuality evaluation, significant challenges remain when employing them as the reward function in online RL.

First, most of these methods focus more on factual precision and lack a reliable way to calculate recall given the difficulty to find all possible relevant facts for a given question. As a

result, it is possible to hack the reward by reducing the detail level in the responses. Indeed, as seen in many previous studies on factuality alignment, optimizing towards factual precision tends to lead to shorter responses and reduced detail level[4].

Furthermore, even if we take the answer detail level into account in the reward design, the reward remains susceptible to spurious inflation. In particular, as the evaluation focuses solely on the factuality of the response, it does not measure the relevance or helpfulness of the answer. It is hence possible to generate a highly detailed and factual response that is less relevant or, in extreme cases, entirely tangential to the question. For instance, in our early experiments, we observed a case where the model was asked, "Who is Leon Wildes?" and it responded with, "Leon Wildes is an immigration attorney. While specific details about his work are not readily available, I can provide more information on immigration law," followed by a large number of correct facts about immigration law. This resulted in a very detailed answer with high factual precision, despite the lack of relevance to the original question.

Last but not least, existing long-form factuality evaluation methods are complex and computationally expensive, making them too slow for real-time reward calculation in online RL. For example, VeriScore, a more recent approach that made several improvements over FactScore and SAFE, but can require several minutes to verify a single response.

In the remainder of the section, we first outline our reward function design, explaining how it mitigates the various ways of hacking the reward as mentioned above. We then briefly describe how we implement a more scalable version of VeriScore, making it suitable for online RL.

### 3.1.1. REWARD FUNCTION

Our overall reward consists of three parts: factual precision ($\mathcal{R}_{fact}$), response detail level ($\mathcal{R}_{dtl}$), and answer relevance ($\mathcal{R}_{rel}$). We define this as follows:

$$\mathcal{R}(y|x) = \begin{cases} -1.0, & \text{if } y \text{ is malformed} \\ \begin{aligned} &\mathcal{R}_{fact} + \lambda \cdot \mathcal{R}_{dtl} + \mu \cdot \mathcal{R}_{rel} \quad \text{otherwise} \\ &= \frac{F}{T+1} + \lambda \cdot \log(1+F) \\ &\quad + \mu \cdot \mathbb{1}(y_{ans} \succ y_{ref}), \end{aligned} \end{cases}$$

(2)

The response $y$ is deemed malformed if it does not follow the

---

[4]There is a nuanced difference between response length and detail level, which is measured by the number of correct facts in the response in this work. It is possible to inflate the response length by generating additional non-factual claims (hallucinations), making it unsuitable as a metric for answer detail level.

format of `<think>` $y_{cot}$ `</think>` `<answer>` $y_{ans}$ `</answer>`. For a given response $y$ to the question $x$, we run VeriScore on $y_{ans}$ to obtain the number of *factual claims* $F$ and *total claims* $T$.

- The first term $\frac{F}{T+1}$ is the *smoothed* factual precision (to avoid zero division).

- The second term $\log(1 + F)$ captures the detail level, which is discounted with a $\log$ factor.

- In the third term, we employ an LLM to check the general quality and relevance of the response $y_{ans}$ compared to the response generated by a reference model $y_{ref}$, and $\mathbb{1}$ is the indicator function which returns 1.0 if $y$ is better than $y_{ref}$ and 0.0 otherwise. In practice, we prompt the base model to first generate its own response to the question, and then compare it with the given response $y_{ans}$ and return a final verdict on which is better. While alternative prompt designs exist, we chose this specific approach because it provides a direct, principled binary signal that effectively tracks whether our RL training degrades the helpfulness of the base model, serving as a reliable guardrail. (The full prompt and more details can be found in Appendix G.)

The hyperparameters $\lambda$ and $\mu$ control the weights of the answer detail factor and the answer quality factor, respectively.

### 3.1.2. SCALABLE VERISCORE IMPLEMENTATION

The original VeriScore implementation is largely sequential, and we optimize efficiency by parallelizing as many operations as possible. For claim extraction, instead of processing sentence by sentence, we collect the LLM requests for all sentences and send them in a batch to an LLM inference engine. Similarly, the verification of all extracted claims can also be parallelized and sent in batch. For searching evidence documents using Google Search via the Serper API, we utilize non-blocking asynchronous API calls, which significantly improved the evidence search speed.

For LLM inference, we leverage the *Matrix* (Wang et al., 2025) library, a fast and scalable LLM inference engine based on vLLM (Kwon et al., 2023), which supports serving multiple replicas of an LLM to improve throughput and handles auto-scaling and load-balancing. We set up a pool of Llama-3.3-70B-Instruct workers via matrix for both claim extraction and verification, where we can send batched asynchronous LLM inference requests to the matrix server to maximize parallelization and throughput. On average, it takes less than 5 seconds to verify a response in our implementation compared to 2 minutes in the original VeriScore implementation. Note that we intentionally use our scalable VeriScore implementation solely during

training while utilizing the standard, original VeriScore for final evaluation. This design choice is twofold: it keeps our final evaluation consistent with the literature, and it serves as partial evidence that our model is not simply overfitting to a specific VeriScore judge.

### 3.2. Group Relative Policy Optimization (GRPO)

We adopt Group Relative Policy Optimization (GRPO, Shao et al., 2024) as the online RL algorithm to optimize for our reward function $\mathcal{R}(y|x)$, following the modified objective in Dr. GRPO (Liu et al., 2025). In particular, GRPO samples a group of responses $G = \{y^1, \ldots, y^N\}$ for a given prompt $x$, and computes a relative advantage of each response by $A(y^i|x) = \mathcal{R}(y^i|x) - \Sigma_{y_j \in G}\mathcal{R}(y_j|x)/N$. GRPO then optimizes the following loss function (with an additional KL divergence term omitted for brevity):

$$\mathcal{L}_{\text{GRPO}} = -\mathbb{E}_{G \sim \pi_{\theta_{\text{old}}}}\left[\sum_{y^i \in G}\sum_t \min\left\{\frac{\pi_\theta(y_t \mid x, y_{<t})}{\pi_{\theta_{\text{old}}}(y_t \mid x, y_{<t})}A(y^i),\right.\right.$$
$$\left.\left.\text{clip}_\epsilon\left(\frac{\pi_\theta(y_t \mid x, y_{<t})}{\pi_{\theta_{\text{old}}}(y_t \mid x, y_{<t})}\right)A(y^i)\right\}\right]$$
$$(3)$$

where we only run one inner step of optimization per each training step.

## 4. Experiments

### 4.1. Datasets and Evaluation Metrics

We choose a diverse set of six long-form factuality datasets for evaluation, where responses are content-rich and open-ended: LongFact (Wei et al., 2024), FAVA (Mishra et al., 2024), AlpacaFact (Dubois et al., 2024), Biography (Min et al., 2023), FactBench (Bayat et al., 2024), and Factory (Chen et al., 2025a). More details about the datasets can be found in Appendix A.

As discussed in Section 3, there are multiple automatic long-form factuality evaluation methods, such as FactScore (Min et al., 2023), SAFE (Wei et al., 2024), and VeriScore (Song et al., 2024). Previous work (Chen et al., 2025b) has shown VeriScore's superiority over previous methods, as it focuses on extracting more sensible *verifiable* claims and uses Google Search instead of Wikipedia as the knowledge source. As a result, VeriScore can be applied to a wider range of topics.

We report *factual precision* (prec. $= F/T$) and *detail level* (dtl. $= F$), where $F$ and $T$ are the number of supported and total claims returned by VeriScore, respectively. We choose to directly report the number of supported claims as a measurement for the comprehensiveness of a response instead of the recall metric proposed in VeriScore. The recall was calculated by dividing $F$ by a global constant $K$, which rep-

*Table 1.* Factual Reasoning results on six long-form factuality benchmarks. Reasoning models are indicated with a background color. Precision (Prec.) is calculated by dividing the number of correct facts (supported claims) by the total number of facts generated in a model response. Detail level (Dtl.) is measured by the number of supported claims in the response. The win rate (WR) is calculated with respect to the non-reasoning base model for each model.

| | LongFact | | | FAVA | | | AlpacaFact | | | Biography | | | Factory-H | | | FactBench-H | | | Average (↑) | | |
|---|---|---|---|---|---|---|---|---|---|---|---|---|---|---|---|---|---|---|---|---|---|
| | Pre. | Dtl. | WR | Pre. | Dtl. | WR | Pre. | Dtl. | WR | Pre. | Dtl. | WR | Pre. | Dtl. | WR | Pre. | Dtl. | WR | Pre. | Dtl. | WR |
| *Existing Reasoning Models* | | | | | | | | | | | | | | | | | | | | | |
| Qwen 2.5 32B | 73.6 | 33.8 | - | 57.8 | 25.2 | - | 65.7 | 27.4 | - | 26.5 | 10.1 | - | 22.6 | 8.9 | - | 62.2 | 26.1 | - | 51.4 | 21.9 | - |
| QwQ 32B | 56.7 | 42.8 | 69.4 | 44.3 | 34.0 | 65.5 | 53.1 | 36.3 | 71.1 | 15.5 | 10.7 | 51.2 | 13.9 | 8.8 | 75.7 | 46.1 | 31.5 | 75.4 | 38.3 | 27.3 | 68.0 |
| DeepSeek V3 671B | 75.0 | **43.1** | - | 63.0 | **36.6** | - | 66.9 | 35.2 | - | 39.8 | **24.0** | - | 22.3 | 10.9 | - | 62.9 | 34.2 | - | 55.0 | 30.7 | - |
| DeepSeek R1 671B | 65.3 | 42.1 | 57.4 | 50.8 | 32.2 | 52.6 | 55.5 | 30.0 | 50.9 | 31.8 | 19.0 | 60.3 | 14.8 | 8.0 | 54.1 | 50.1 | 30.5 | 54.7 | 44.7 | 27.0 | 55.0 |
| Llama-3.1-8B-Instruct | 60.0 | 36.8 | - | 48.3 | 30.7 | - | 60.2 | 30.7 | - | 25.9 | 10.3 | - | 22.2 | 7.8 | - | 53.3 | 28.3 | - | 45.0 | 23.5 | - |
| *Offline Factual Reasoning Training* | | | | | | | | | | | | | | | | | | | | | |
| 8B SFT | 75.1 | 26.2 | 55.7 | 62.1 | 18.5 | 49.0 | 70.1 | 22.9 | 43.1 | 36.5 | 8.9 | 56.5 | 26.6 | 7.1 | **54.9** | 65.1 | 21.3 | 47.0 | 55.9 | 17.5 | 51.0 |
| 8B SFT + DPO | **79.7** | 32.4 | 47.2 | **70.5** | 26.6 | 33.7 | 79.2 | 33.9 | 32.5 | 46.6 | 12.1 | 48.3 | 50.9 | 16.4 | 30.1 | **79.7** | 33.0 | 35.0 | 67.8 | 25.7 | 37.8 |
| *Online Factual Reasoning Training* | | | | | | | | | | | | | | | | | | | | | |
| 8B SFT + GRPO | 79.4 | 37.2 | **65.2** | 70.3 | 30.8 | **51.3** | **79.7** | 36.9 | 50.4 | **47.1** | 14.8 | **59.8** | 54.8 | 20.2 | 47.1 | 77.1 | **34.3** | **52.8** | **68.1** | 29.0 | **54.4** |

resents the median number of extracted claims per dataset, serving as an estimate of how many claims are expected to achieve a perfect recall. We believe this approximation does not accurately reflect the true upper bound of the expected number of claims, and it can even result in a recall greater than 1. Therefore, it is more informative to directly report $F$, the number of supported facts, rather than normalizing it by a semi-arbitrary constant.

In addition to factuality evaluation, we also evaluate the helpfulness of a response to ensure that a model does not produce irrelevant but factually correct answers. Following (Lin et al., 2024) we use the AlpacaEval (Dubois et al., 2024) prompt template with a GPT-4o judge to compare responses from the target model and the base model across each of the six datasets, which calculates the win rate based on their instruction-following ability. The discrepancy between using a weaker model (Llama-3.1-8B-Instruct) for the training reward and a frontier model (GPT-4o) for final evaluation is deliberate. First, it ensures the quality of the final evaluation by leveraging a state-of-the-art model. Additionally, demonstrating performance improvements under a GPT-4o evaluator when trained with a Llama-3.1-8B judge suggests that our proposed method is relatively stable and generalizable across different choices of LLM evaluators.

### 4.2. Evaluating Existing Reasoning Models on Long-Form Factuality

In the top half of Table 1, we conduct an extensive empirical analysis of the performance of two popular Reasoning LLMs, namely DeepSeek-R1 (DeepSeek-AI, 2025b) and QwQ-32B (Qwen-Team, 2025), on the six long-form factuality benchmarks described in Section 4.1. Compared with their non-reasoning counterparts, DeepSeek-V3 (DeepSeek-

AI, 2025a) and Qwen-2.5-32B (Yang et al., 2024), respectively, both R-LLMs exhibit significantly more hallucinations on *all* datasets, with double-digit decreases in factual precision on average.

Interestingly, while QwQ appears to trade lower precision (51.4 → 38.3) with a higher detail level (21.9 → 27.3), DeepSeek-R1 has a higher hallucination rate while producing less details than DeepSeek-V3. Overall, it is observed that existing reasoning models, despite making substantial advances to complex reasoning tasks such as mathematics and programming, fail to improve factuality and exacerbate the hallucination issue.

### 4.3. Factual Reasoning Results

Our main results can be found in the bottom half of Table 1. We use Llama-3.1-8B-Instruct as the base model in our training, and compare our factual reasoning models with it in terms of factuality and helpfulness as outlined in Section 4.1.

We first show that offline RL approaches, as seen in previous work on factuality alignment (Lin et al., 2024), can improve factual precision but ultimately decrease the overall quality in responses. In particular, the SFT model brings an average of 10.9 points increase in precision, but also reduces the detail level by more than 25%. In contrast, the SFT + DPO model is able to further enhance the precision, to +22.8 points on average over the base model, while also maintaining the same or higher level of detail on 4 out of the 6 datasets. However, the average win rate drops significantly below 50% to 37.8%, indicating a drastic degradation in response quality compared to the base model.

On the other hand, our online RL approach (SFT + GRPO) manages to boost factual precision and detail level with-

out compromising the overall relevance or quality of the answers, thanks to our comprehensive reward function and on-policy optimization. It achieves an average of 68.1% precision, 23.1 points over the base model, and a 23% relative increase in detail level. Furthermore, its 54.4% win rate vs. the base model demonstrates that our factual reasoning model produces meaningful and pertinent responses while substantially reducing hallucination. As we shall discuss in Section 5.1, it is also possible to achieve different trade-off points between precision and detail, by changing the weights of the various components in our reward function.

# 5. Ablations and Analysis

## 5.1. Reward Design Ablations

In this section, we analyze the impact of the various components in our reward function. Starting from $\mathcal{R}_{fact}$ alone (row 2), where the model optimizes solely for factual precision, we observe that the precision indeed improves by 24.7 percentage points, and the detail level (#sup) also increases, see Table 2. However, the 43.5% win rate suggests the presence of reward hacking, with some facts in the response being less relevant to the question. This point is further illustrated when $\mathcal{R}_{dtl}$ is added (row 3), which counterintuitively leads to additional improvements in both precision and detail level. Nevertheless, the win rate further dropped to 36.9%, and a closer examination reveals that the precision and detail level are falsely inflated by including more general (factually correct) statements that are less pertinent to the question. Combining $\mathcal{R}_{fact}$ and $\mathcal{R}_{rel}$ (row 4), on the other hand, successfully mitigates the reward hacking issue, and achieves a 54.4% win rate over the base model while increasing precision by 23.1 points and detail level by 23%. When further adding $\mathcal{R}_{dtl}$ to combine all three components (row 5&6), a controlled trade-off can be achieved between the precision and detail level. Compared to row 4, row 5 attains a 13% detail level increase at the expense of 1 point drop in precision and a slight decrease in the win rate, which still remains above 50%. With $\lambda = 0.1$ (row 6), which assigns a much higher weight to the detail level reward, the model generates 88% more factual claims than the base model, while having a 22 points higher precision. However, the lowered win rate of 45.7% shows that reward hacking resurfaces with the increased emphasis on detail level. Nonetheless, it is less severe compared to when solely optimizing for $\mathcal{R}_{fact}$, proving the utility of the relevance reward $\mathcal{R}_{rel}$.

We conclude that both row 3 and 4 are reasonable choices in practice, depending on whether one prefers more accurate answers that are directly relevant to the question, or more detailed responses that sometimes include additional related information. In Table 1, we report $\lambda = 0, \mu = 0.1$ as our main model, yet the main conclusions remain the same with the $\lambda = 0.01, \mu = 0.1$ model.

## 5.2. Analysis of the Factual Reasoning CoT Traces

Figure 2 illustrates the trajectory of how the lengths of the CoT reasoning traces and the answers change during GRPO training. We observe that both lengths increase sharply in the early stages of training, then plateau and fluctuate thereafter. One hypothesis is that the model initially learns to produce more detailed responses, and subsequently shifts to fine-tune its reasoning strategies and answers to further improve the factual precision and answer relevance. Figure 3 shows the length distribution of the CoT reasoning chains from the trained model, calculated on 4k training prompts.

We also employ Llama-3.1-70B-Instruct to identify the "meta-reasoning" strategies in the CoT thinking process, such as self-verification, backtracking, summarization, etc., using a similar prompt adapted from (Li et al., 2025) (Full prompt in Appendix H). The top 20 most frequently used strategies are shown in Figure 4. We find that the factual reasoning strategies exhibit noticeable differences from those found in existing reasoning datasets (Li et al., 2025, Figure 3). The reasoning strategies in the NATURALTHOUGHTS dataset are primarily focused on solving math or coding problems, such as self-verification, exploration, calculation, and backtracking. Conversely, our factual reasoning model employs a more diverse set of strategies that are better suited for fact-seeking questions, utilizing techniques such as synthesis, summarization, explanation, definition, and comparison.

# 6. Related Work

**Reasoning Large Language Models** Recent advances in Reasoning Large Language Models (R-LLMs) have been driven by the application of reinforcement learning (RL) techniques at scale that enable Long Chain-of-Thought (Long CoT) reasoning, pioneered by OpenAI's o1 (OpenAI, 2024) and further democratized by the open-source DeepSeek-R1 model (DeepSeek-AI, 2025b), where the RLVR approach of using verifiable rewards is the crucial driver (Luong et al., 2024; Pang et al., 2024; Lambert et al., 2025). Such Long CoT reasoning traces allow a model to "think longer" and iteratively self-refine its reasoning process before producing an answer. This approach significantly improves the LLM's capabilities in complex reasoning tasks such as mathematics and programming, and has sparked a long line of subsequent research (Chen et al., 2025c). For example, QwQ highlights the potential of smaller R-LLMs by applying RLVR to a 32B model. LIMO (Ye et al., 2025) and s1 (Muennighoff et al., 2025) challenge the need for massive training data and show that competitive performance can be achieved with only a fraction of distilled SFT data. Most existing work on R-LLMs, however, focuses on reasoning

*Table 2.* Ablation results on the reward function design. Factuality, Detail, and Relevance evaluations are reported as the averages on the six benchmarks.

| | $\mathcal{R}(y\|x)$ | $\lambda$ | $\mu$ | Factuality & Detail | | | Relevance |
| | | | | Pre. | #sup. | #unsup. | Win Rate |
|---|---|---|---|---|---|---|---|
| 1 | Llama-3.1-8B-Instruct | - | - | 45.0 | 23.5 | 22.1 | 50.0 |
| 2 | $\mathcal{R}_{fact}$ | 0 | 0 | 69.7 | 30.0 | 10.1 | 43.5 |
| 3 | $\mathcal{R}_{fact}$ & $\mathcal{R}_{dtl}$ | 0.01 | 0 | 74.5 | 38.3 | 10.5 | 36.9 |
| 4 | $\mathcal{R}_{fact}$ & $\mathcal{R}_{rel}$ | 0 | 0.1 | 68.1 | 29.0 | 10.8 | 54.4 |
| 5 | $\mathcal{R}_{fact}$ & $\mathcal{R}_{dtl}$ & $\mathcal{R}_{rel}$ | 0.01 | 0.1 | 67.1 | 32.8 | 12.8 | 51.8 |
| 6 | $\mathcal{R}_{fact}$ & $\mathcal{R}_{dtl}$ & $\mathcal{R}_{rel}$ | 0.1 | 0.1 | 67.0 | 44.2 | 17.0 | 45.7 |

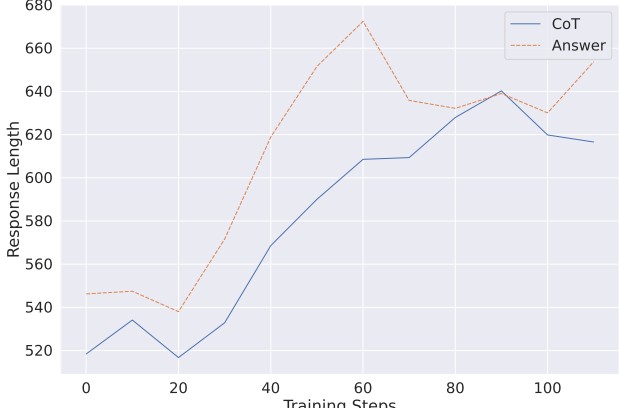

*Figure 2.* Length trajectory of the CoT reasoning traces during Factual Reasoning training.

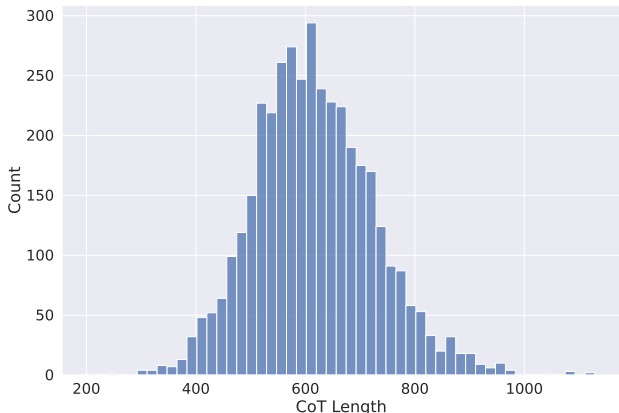

*Figure 3.* CoT length distribution on 3920 training prompts.

tasks and often overlooks the hallucination issue that arises in these Long CoT models.

**LLM Post-Training for Factuality** Post-training (alignment) techniques have been proposed to enhance the factuality of LLMs (Tian et al., 2023; Lin et al., 2024; Zhang et al., 2024), mostly focused on supervised finetuning (SFT) and offline RL approaches such as DPO (Rafailov et al., 2023). These works have shown that it is important to optimize factuality of self-generated responses during post-training (i.e., be on-policy), rather than finetuning on unfamiliar knowledge (e.g, reference facts) which may actually increase hallucinations (Kang et al., 2025; Gekhman et al., 2024; Ghosal et al., 2024; Zhang et al., 2024). A recent work (Peng et al., 2025) proposes a pairwise reward model that combines a standard reward model reflecting human preference with additional verification agents, such as FActScore, and applies it in DPO training.

The factuality aspect of R-LLMs has been under-explored until very recently. Concurrent papers start to study RL methods for improving factuality in R-LLMs (Li & Ng, 2025; Ren et al., 2025), but they focus on **short-form factoid questions** where the answers can be easily verified against the ground truth. Our work, in contrast, considers the more general long-form factuality problem, where it is much more challenging to design an effective and efficient

reward function.

## 7. Conclusion

In this work, we investigate the factuality issue in Reasoning LLMs, revealing that existing models have substantially more hallucinations in long-form responses, and propose a new online RL approach for learning more accurate *factual reasoning*. In particular, we develop a new reward function that combines VeriScore, an automatic long-form factuality evaluation method, and an LLM judge to comprehensively assess a response in terms of its factual precision, response detail level, and answer relevance. Evaluated on six long-form factuality benchmarks, our factual reasoning model reduces the hallucination rate by 23.1 percentage points and increases the response detail level by more than 20%. Furthermore, it maintains a >50% AlpacaEval win rate over the base model, indicating no degradation in the model's general instruction-following capabilities.

For future work, it would be intriguing to apply factual reasoning in the agentic setting where the model has access to tools such as a search engine. This setting could unlock numerous new reasoning strategies for the model to improve the factuality in the response. For example, the model may reason about details that it is uncertain about, issue relevant search queries to find the missing knowledge, and inspect

the search results for relevant information. Recent work (Jin et al., 2025) has studied agentic factual reasoning in **short-form** factuality, where the model is trained on simple factoid questions with short answers that can be easily verified in the reward calculation. It has been less explored in the more general and challenging long-form factuality domain, and we believe our online RL approach with the long-form factuality rewards can be extended to the agentic setting to facilitate learning agentic factual reasoning in long-form responses.

## Limitations

While our proposed online RL framework demonstrates significant improvements in long-form factuality, several limitations remain. First, our evaluation and training pipelines rely heavily on automatic evaluators (e.g. VeriScore) and LLM-as-a-judge frameworks. Developing accurate metrics for long-form open-ended generation remains an open challenge. Second, due to the prohibitive cost of human evaluation on long-form factual verification (estimated at $4 per response in Min et al., 2023), our study lacks human validation. Finally, there is an inherent risk that optimizing our models against a specific verification stack, such as VeriScore, might bias the model's behavior toward the specific preferences or quirks of that stack, although we attempt to mitigate this by using different models for training and evaluation.

## Reproducibility Statement

We provide additional implementation details in Appendix B, and all the LLM prompts used in our experiments can be found in the Appendices. To further facilitate reproducibility, we release the following artifacts to the public: Our training code, the scalable VeriScore implementation, as well as our training data.

**Training Code** https://github.com/facebookresearch/factual_reasoning

**ScalableVeriScore** https://github.com/facebookresearch/ScalableVeriScore

**Training Data** https://huggingface.co/datasets/facebook/factual_reasoning

## Impact Statement

This paper presents work whose goal is to advance the field of machine learning. There are many potential societal consequences of our work, none of which we feel must be specifically highlighted here.

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

# A. Evaluation Datasets

We choose a diverse set of six long-form factuality datasets for evaluation, where responses are content-rich and open-ended.

**LongFact** (Wei et al., 2024) contains questions of which the intended responses consist of at least several paragraphs. It was created by prompting GPT-4 to generate questions regarding a specific concept or object within a given topic. In our experiments, we use the 250 prompts from the LongFact-Objects dataset, selected by the original authors.

**FAVA** (Mishra et al., 2024) is a fine-grained hallucination benchmark with 200 information-seeking queries that require factual knowledge to give accurate long-form answers from multiple sources. Following (Lin et al., 2024), we select 141 prompts from this dataset in our experiments.

**AlpacaFact** is a subset of 241 fact-seeking instructions from the AlpacaFarm (Dubois et al., 2024) dataset which has 805 real-world instructions from various users, selected by (Lin et al., 2024).

**Biography** (Min et al., 2023) has 183 questions in the form of "Tell me a bio of [`PERSON NAME`]" with names selected from Wikipedia.

**FactBench** (Bayat et al., 2024) applies automatic filtering to a select subset of LMSYS-Chat-1M prompts, using responses from various LLMs based on hallucination scores, to obtain a set of challenging prompts in terms of factuality. We evaluate on its most challenging tier, FactBench-Hard with 532 questions.

**Factory** (Chen et al., 2025a) is a new long-form factuality benchmark with human-verified challenging prompts where frontier LLMs are only achieving approximately 40% factual precision.

# B. Implementation Details

We train all models using the `fairseq2` library (Balioglu et al., 2023), which supports spawning a set of `vllm` (Kwon et al., 2023) inference workers for policy model, reference model, and LLM-as-a-judge reward model.

For calculating the VeriScore reward, we set up a matrix cluster of 8 Llama-3.3-70B-Instruct workers on 32 NVIDIA H100 GPUs for claim extraction and verification. We create an API webserver using Quart[5] so that requests can be sent to our VeriScore server remotely from the training workers. For evaluation, we use the finetuned claim extractor[6] and verifier[7] in the original VeriScore to maintain consistency with the existing literature, which is run on a single H100 GPU. Furthermore, we implemented data sharding to speed up offline evaluation runs by leveraging multiple GPUs.

Our SFT run is trained on 8 H100 GPUs for 1 epoch with a per-GPU batch size of 4 and a learning rate of 5.5e-6. The offline DPO run is trained on 16 H100 GPUs for 1 epoch, with a batch size of 1 and a learning rate of 1e-6. For online GRPO, we use 32 H100 training workers and 8 H100 inference workers. We use 4 rollouts per prompt, and the model is trained for 1 epoch with a batch size of 1 and learning rate of 1e-6. More details can be found in Lanchantin et al. (2025) on the online RL implementation in `fairseq2`.

# C. Analysis of the Factual Reasoning CoT Traces

Figure 4 shows the top 20 most frequently used strategies by our factual reasoning model, as discussed in Section 5.2

# D. Example Model Output

Tables 3 and 4 present example responses to the question "What is the Bellevue Hospital Center?" (from the non-test split of LongFact), comparing the output from the base model and our factual reasoning model. As shown in Table 3, the base model tends to produce many plausible yet factually incorrect details such as dates, numbers, the specific people, making the answer disinformative and potentially misleading. On the other hand, our factual reasoning model (Table 4) leverages its CoT thinking process to double check details such as dates and exact locations, while abstaining from generating facts that the model is not certain about. For instance, while both models correctly identify Bellevue Hospital Center's notable history in psychiatric care, our model suppresses hallucinated details—such as the exact year of establishment and the name of

---

[5]https://quart.palletsprojects.com/en/latest/
[6]https://huggingface.co/SYX/mistral_based_claim_extractor
[7]https://huggingface.co/SYX/llama3_based_claim_verifier

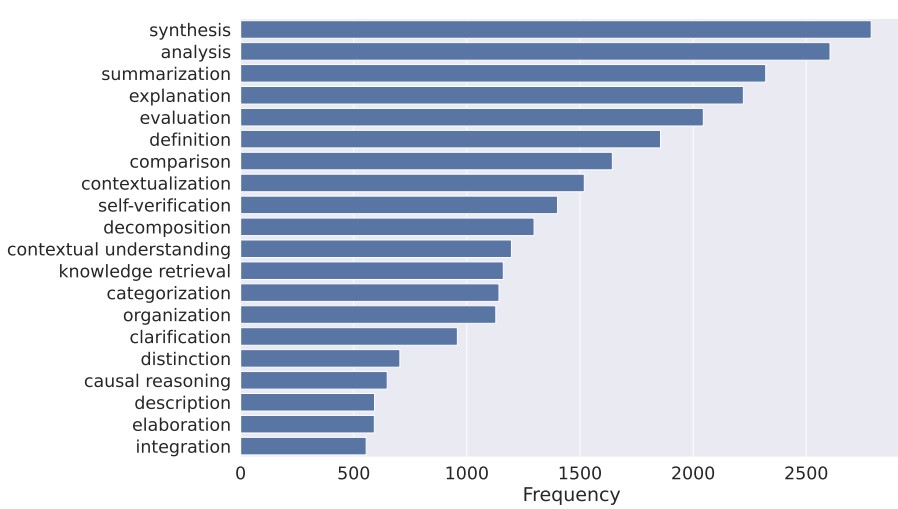

*Figure 4.* Top 20 commonly used reasoning strategies based on 3920 training prompts.

the founder—that appear in the base model's response. Similarly, both models mentions that its Emergency Department is one of the busiest in New York City, but the base model also generates a hallucinated number of annual visits. Our factual reasoning training was also able to surface details that the model knew about but was omitted in the base model's response, such as its exact address and a notable event of the first use of insulin coma therapy in history.

## E. Prompt for Generating Synthetic Training Prompts

Figure 5 illustrates the Llama 4 prompt for generating the synthetic fact-seeking questions used in our training. Figure 6 shows randomly sampled examples from the generated synthetic training prompts.

---

**Synthetic fact-seeking prompt generation**

```
You have a task to generate a prompt for a language model that requires factual
knowledge for a high-quality answer.  Most of such prompts involve named entities
such as personas from history or historical events.  We need to go beyond such
prompts and cover commonly known knowledge beyond named entities.

Examples below are factual prompts wrapped in <factual_prompt> tags:

<factual_prompt>fp1</factual_prompt>
<factual_prompt>fp2</factual_prompt>

Examples below are prompts that are more diverse and may not require factual
knowledge, wrapped in <prompt> tags:

<prompt>p1</prompt>
<prompt>p2</prompt>
<prompt>p3</prompt>
<prompt>p4</prompt>

Now please generate a new prompt that will require factual knowledge for an answer
(as in the first group), but it has to be more connected to real world example (as in
the second group).

New prompt in your response has to be wrapped in <new_prompt> tags.
```

*Figure 5.* LLM prompt template for generating synthetic fact-seeking prompts, where seed prompts fp1, fp2 and p1-p4 should be provided. We use real-world diverse prompts from WildChat (Zhao et al., 2024) to sample p1-p4 and a set of fact-seeking prompts from the non-test split of LongFact (Wei et al., 2024) to sample fp1 and fp2.

---

**Examples of synthetic training prompts**

```
How do saltwater intrusion barriers work in protecting freshwater aquifers near
coastal areas?

What is the purpose of a DNS resolver in a computer network?

Why do some materials conduct electricity better than others?

What are the main differences between OLED and QLED display technologies?

How does a typical day in the life of a professional involved in beekeeping look like,
and what are some of the key tasks they perform?

How do anti-reflective coatings on lenses work, and what are the principles behind
their design?

How does a typical credit card's interest calculation work?

What are the implications of de-emulsification in the context of oil spills in marine
environments?
```

*Figure 6.* LLM prompt for synthetic fact-seeking prompt generation.

## F. Few-shot Prompt for Generating Long CoT Responses

Figure 7 and 8 depicts the manually written few-shot prompt to create seed Long CoT data for SFT and DPO.

## G. Prompt for the LLM Judge in the Answer Relevance Reward

Figure 9 shows the prompt we use to judge the general answer quality and relevance to the question, used in our answer relevance reward $\mathcal{R}_{rel}$. We measure any potential degradation in quality over the base model, we adopt a pairwise judgment method which compares the rollout with a reference response generated by the base model to produce a binary judgment on which response is better. For simplicity, we leverage the base model as the judge which enables us to use a single prompt to first ask the LLM to generate its own response, and then act as an impartial judge to compare its response with the given one. Using the base model as the answer relevance judge provides a simple, direct, and principled **binary signal** that effectively tracks whether the policy model maintains the same answer quality as the base model, thereby preventing degradation in helpfulness. In practice, we find this approach to work reasonably well, so we did not further explore the use of more powerful LLMs as the judge.

## H. Prompt for LLM-based Meta-Reasoning Strategy Identification

We follow a similar approach by (Li et al., 2025) to ask an LLM to analyze the reasoning chains generated by our model, and summarize what "meta-reasoning" strategies are used, with the full prompt shown in Figure 10. In addition, we also ask the LLM to provide an overall helpfulness score, on a scale of 0 to 10, to assess the extent to which the reasoning chain helps the model produce a more factual, detailed, and relevant answer. However, we notice that the LLM generates a score of 9 / 10 for the vast majority of the questions, and 8 / 10 for the remaining. While this affirms the overall helpfulness of the factual reasoning traces, it is also possible that the LLM lacks a reliable way to accurately gauge the helpfulness. We hence did not include this result in the main paper.

*Table 3.* Example response from the base model. See Table 4 for the response from our factual reasoning model. Texts in red indicate factual errors / hallucinations.

---

**Question:** What is the Bellevue Hospital Center?

---

**Response from Base Model (Llama-3.1-8B-Instruct)**

---

The Bellevue Hospital Center, officially known as Bellevue Hospital Center, is a 1,600-bed public hospital located in New York City, specifically in the East Village neighborhood of Manhattan. It is one of the oldest continuously operating hospitals in the United States, with a rich history dating back to 1738.

**History:**

Bellevue Hospital was founded by John Watts, a British physician and philanthropist, on April 26, 1738, as the "New York Hospital" in a small building on the corner of Garden Street (now Cherry Street) and the East River. The hospital's name "Bellevue" was derived from the French term "beau vue," meaning "beautiful view." In 1816, the hospital was relocated to its current site on First Avenue and 27th Street, and it was renamed Bellevue Hospital.

**Notable Features and Services:**

1. **Psychiatric Services:** Bellevue Hospital has a long history of providing psychiatric care, dating back to 1839. The hospital's psychiatric ward was established by Dr. Samuel B. Woodward, who is often credited with pioneering the concept of asylums. In 1902, the hospital's psychiatric services were expanded to include a new building designed by architects Carrère and Hastings.
2. **Emergency Department:** The hospital's emergency department is one of the busiest in the country, with over 170,000 visits annually.
3. **Trauma Center:** Bellevue Hospital is a Level I Trauma Center, providing emergency care for critically injured patients.
4. **Forensic Services:** The hospital's psychiatric services also include a forensic unit, where patients who have committed crimes are evaluated and treated.

**Notable Events and Incidents:**

1. **The Bellevue Insane Asylum Riot (1839):** On August 10, 1839, a riot broke out at Bellevue's psychiatric ward, resulting in the deaths of 13 patients and several staff members.
2. **The Bellevue Hospital Fire (1865):** A fire broke out on May 6, 1865, at Bellevue Hospital, killing 20 people and injuring many others.
3. **The 1918 Spanish Flu Pandemic:** Bellevue Hospital was severely impacted by the 1918 Spanish Flu pandemic, with over 1,000 patients dying in a single week.
4. **The 9/11 Attacks:** Bellevue Hospital treated over 1,600 patients from the 9/11 attacks, including many with severe injuries.

**Notable People:**

1. **Dr. Samuel B. Woodward (1798-1850):** A physician and psychiatrist who established Bellevue's psychiatric ward and pioneered the concept of asylums.
2. **Dr. Abraham Flexner (1866-1959):** A medical educator and hospital administrator who served as the hospital's superintendent from 1908 to 1913.
3. **Dr. Willard Van Orman Quine (1908-2000):** A philosopher and logician who worked at Bellevue Hospital as a psychiatrist.
4. **Dr. Norman C. Nelson (1925-2011):** A physician and hospital administrator who served as the hospital's president and CEO from 1974 to 1983.

**Statistics:**

1. **Annual Budget:** Over $2.5 billion
2. **Number of Employees:** Over 6,000
3. **Number of Patients:** Over 100,000 per year
4. **Number of Beds:** 1,600

**Awards and Recognition:**

1. **Magnet Hospital Designation:** Bellevue Hospital has been designated as a Magnet hospital by the American Nurses Credentialing Center (ANCC) for excellence in nursing care.
2. **Leapfrog Group Top Hospital:** Bellevue Hospital has been recognized as a Top Hospital by the Leapfrog Group, a nonprofit organization that evaluates hospital safety and quality.

Bellevue Hospital Center is a vital part of New York City's healthcare system, providing essential services to the community and serving as a teaching hospital for the New York University School of Medicine.

---

*Table 4.* Example response from our factual reasoning model (SFT+GRPO). See Table 3 for the response from the base model. Texts in red indicate factual errors / hallucinations.

---

**Question:** What is the Bellevue Hospital Center?

---

**Response from the Factual Reasoning Model (SFT + GRPO)**

---

```
<think>
```
The user is asking about the Bellevue Hospital Center. To provide accurate information, I should start by confirming what Bellevue Hospital is. It's a hospital in New York City, specifically located in Manhattan. The full name is Bellevue Hospital Center, and it's one of the oldest continuously operating hospitals in the United States.

To ensure the accuracy of the information, I'll verify its history. Bellevue Hospital was founded in 1736 as a almshouse, which was a type of institution that provided care for the poor. Over time, it evolved into a hospital. The hospital is associated with New York University (NYU) and is a major teaching hospital, which means it plays an important role in medical education and research.

Let me check the exact location – it's on First Avenue in Manhattan. The address is 462 First Avenue, New York, NY 10016. This information should be correct, as I've verified it before.

Now, about its services and departments. Bellevue Hospital is known for its emergency department, which is one of the busiest in the city. It's also a major trauma center. I should note that it has a wide range of medical specialties, including psychiatry, which is significant given its historical use as a psychiatric hospital.

Bellevue Hospital has been involved in various notable events and has a rich history, especially in terms of psychiatric care. It was the site of many important developments in psychiatry, including the development of insulin coma therapy for schizophrenia. I should be careful not to mix up historical events, so I'll make sure to keep the details clear.

In terms of specific events, one notable example is the role Bellevue played in the development of modern psychiatric care. It was also a site for the treatment of patients during the COVID-19 pandemic, highlighting its ongoing role in healthcare service.

Lastly, I should check for any recent updates or notable achievements, but the information might not be up-to-date as of my knowledge cutoff. However, Bellevue's history and its current status as a teaching hospital and emergency center are well-documented.

Now, assembling all the elements to present a comprehensive answer that covers history, services, and significance of Bellevue Hospital Center.
```
</think>
<answer>
```
The Bellevue Hospital Center, located at 462 First Avenue, New York, NY 10016, is one of the oldest continuously operating hospitals in the United States. Founded in 1736 as an almshouse, it has a rich history that spans over two and a half centuries, evolving over time to become a comprehensive medical center.

### History
- **Founding and Early Years**: Bellevue Hospital was established in 1736 as a place to care for the poor, reflecting the charitable efforts of the community and the need for medical care for those without resources.
- **Evolution**: Over the years, the institution has undergone significant transformations, adapting to the needs of the city and the advancements in medical science. It became a hospital and expanded its services to include a wide range of medical specialties.
- **Psychiatric Care**: Bellevue has a particularly notable history in psychiatric care, having been a major center for psychiatric treatment and research. This aspect of its history is significant, reflecting the evolution of psychiatric care and treatment approaches.

### Services and Departments
- **Emergency Department**: Bellevue Hospital is known for its emergency department, which is one of the busiest in New York City, indicating its critical role in urgent care.
- **Specialties**: The hospital offers a broad spectrum of medical specialties, contributing to its status as a comprehensive medical center.
- **Teaching Hospital**: Bellevue Hospital is associated with New York University (NYU), serving as a teaching hospital, which underscores its importance in medical education and training.

### Notable Events and Achievements
- **Psychiatric Care Innovations**: Bellevue has been at the forefront of psychiatric care, with significant contributions to the development of treatment approaches, such as insulin coma therapy for schizophrenia, a notable example of its historical impact on psychiatric treatment.
- **COVID-19 Response**: Like many healthcare facilities, Bellevue Hospital played a critical role in responding to the COVID-19 pandemic, highlighting its ongoing relevance in contemporary healthcare.

### Location and Accessibility
- **Address**: The hospital is located at 462 First Avenue, New York, NY 10016, in Manhattan, making it easily accessible to a large population.
- **Accessibility**: As a major medical center, Bellevue is equipped to handle emergency situations and provides a range of services, contributing to the healthcare needs of the community.

### Conclusion
Bellevue Hospital Center, with its long history, comprehensive services, and contributions to medical education and research, remains an important institution in New York City and in the broader medical community, reflecting the evolution of healthcare and the commitment to serving the community's health needs.
```
</answer>
```

**2-shot Prompt for Generating Responses with Factuality-Focused Reasoning Chains**

When a user asks a question, you should first think about the reasoning process in the mind and then provides the user with the answer. The reasoning process and answer are enclosed within <think> </think> and <answer> </answer> tags, respectively, i.e., <think> reasoning process here </think> <answer> answer here </answer>. The thinking process should focus on ensuring the factuality of the final answer. For example, when asked about a certain entity or event, the assistant should first try to recollect all the details it knows about it. Special attention should be paid to areas that are more likely to lead to factual errors, such as dates, names, numbers, etc. When appropriate, double check details to make sure they're factually correct in the thinking process.
Here are some examples:
User: What can you tell me about the Haymarket Riot?
Assistant:
<think>
The user asks about the Haymarket Riot. Let me start by confirming the correct name. It's commonly known as the Haymarket Affair, but can be referred to as the Haymarket Square Riot or Haymarket Riot.
Let me recall all the facts I know about this. It happened in Chicago in 1886 I think. It was part of the labor movement struggles for eight-hour workday. It started as a peaceful rally in support of the striking workers. Then a bomb was thrown, leading to multiple casualties. There were subsequent trials where anarchists were convicted, some executed.
First let me confirm the place and date: Chicago, May 1886. Confirming the exact date: It should be May 4, 1886. Now, let me recall the key events leading up to the riot. I believe it started with a general strike on May 1, 1886, where thousands of workers across the United States walked off their jobs to demand an eight-hour workday. It happened after a May 3 rally at the McCormick Reaper Plant, during which demonstrators were killed. Wait, were there 2 or 6 deaths? I can't remember for sure, so I'll just mention multiple casualties.

The rally in Haymarket Square was organized by anarchist leaders, including August Spies and Albert Parsons, to show support for the striking workers. The event began peacefully, but things took a turn when police arrived to disperse the crowd. That's when the bomb was thrown, killing at least one police officer and injuring many more.
Let me check the details like timing and attendance numbers, since these are prone to errors. I think the rally started at around 7:30pm, but I'm not sure so it's better to leave it out. I do remember, though, the bombing happened at around 10:30pm. The exact attendance is unknown. I remember it was between 1500 and 3000, but there are other estimations as well. I'll need to mention that.
After the bombing, there was widespread outrage and calls for revenge against the labor movement. Eight anarchist leaders were arrested, tried, and convicted of conspiracy and murder. Four of them – August Spies, Albert Parsons, Adolph Fischer, and George Engel – were executed by hanging on November 11, 1887. The others received lengthy prison sentences. Wait, did I get the names correctly? August Spies, Albert Parsons, Adolph Fischer, and George Engel – those were indeed the ones executed.
Also, let me double-check the date of the executions: November 11, 1887, is correct. And double-check the number of executions: four.
Lastly, I'll verify the aftermath and long-term impact of the Haymarket Riot. The trial was criticized as unfair, lack of evidence linking individuals to the bomb. Pardons were later issued. Mention the Haymarket martyrs. Also, note that May 1st, International Workers' Day, stems from this event. Need to avoid inaccuracies.
Alright, assembling all elements now to present a coherent answer. Make sure it's clear, accurate, and covers the essential points the user is likely interested in. Maybe mention the historical significance, the anarchist involvement, the trial's controversy, and the legacy in labor rights and May Day celebrations.

</think>
<answer>
The Haymarket Riot, also known as the Haymarket Affair or Haymarket Square Riot, was a pivotal event in American labor history that took place on May 4, 1886, in Chicago, Illinois. The incident occurred during a labor protest and rally at Haymarket Square, which was part of a larger movement advocating for an eight-hour workday.

Here are the key details surrounding the event:
**Background and context:**
– In the late 19th century, the labor movement was gaining momentum in the United States, with workers demanding better working conditions, higher wages, and reduced working hours.
– The Federation of Organized Trades and Labor Unions (FOTLU), a precursor to the American Federation of Labor (AFL), had been pushing for an eight-hour workday since 1884.
– On May 1, 1886, labor unions across the country organized strikes and demonstrations to mark the beginning of a nationwide campaign for an eight-hour workday.
**The events leading up to the Haymarket Riot:**
– On May 3, 1886, a labor protest was held in Chicago, which turned violent when police fired on a group of strikers at the McCormick Reaper Plant, killing several people.
– In response to the police brutality, labor leaders, including August Spies and Albert Parsons, organized a rally at Haymarket Square for May 4 to protest the police actions and advocate for the eight-hour workday.
**The Haymarket Riot:**
– The rally at Haymarket Square began on May 4, 1886, with less than 3,000 people in attendance (estimations vary).
– The event was peaceful until around 10:30 pm, when a group of police officers arrived at the scene.
– At 10:30 pm, a bomb was thrown into the crowd of police, resulting in police casualties.
– The police responded with gunfire, shooting into the crowd of protesters, killing at least four civilians and injuring many others.

**Aftermath and trial:**
– The incident led to widespread outrage and a massive manhunt for the perpetrators of the bombing.
– Eight anarchist labor leaders, including August Spies, Albert Parsons, and Samuel Fielden, were arrested and charged with murder and conspiracy.
– The trial was highly publicized and politicized, with the prosecution arguing that the defendants were responsible for the bombing, despite a lack of direct evidence.
(TO BE CONTINUED..)

*Figure 7.* LLM prompt for synthetic fact-seeking prompt generation.

**(Cont.) 2-shot Prompt for Generating Responses with Factuality-Focused Reasoning Chains**

```
- In August 1886, the jury delivered a guilty verdict, and seven of the defendants were sentenced to death.
One defendant, Oscar Neebe, was sentenced to 15 years in prison.
- On November 11, 1887, four of the defendants, August Spies, Albert Parsons, Adolph Fischer, and George Engel,
were hanged.
**Legacy:**
- The Haymarket Riot and the subsequent trial were widely condemned by labor unions and socialists, who saw
the event as a miscarriage of justice and an example of government repression.
- The incident led to the establishment of International Workers' Day on May 1st.
- The Haymarket Riot also led to increased public awareness of labor issues and the growth of the labor
movement in the United States.
In 1893, Illinois Governor John Peter Altgeld pardoned the remaining three defendants, citing a lack of
evidence and a biased trial.  The Haymarket Riot remains an important event in American labor history,
symbolizing the struggles of workers for better working conditions and the right to organize.

</answer>
User:  What is the Pech Merle Cave?
Assistant:
<think>
Okay, the user is asking about the Pech Merle Cave.  Let me start by recalling what I know about this.  I
believe it's a prehistoric cave located in France, known for its ancient rock art and human remains.
To provide more accurate information, I'll recall the details I know about this cave.  It's situated in the
Lot department of southwestern France, near the town of Cabrerets.  The cave is one of the few prehistoric
sites in France that still has its original cave paintings intact.  Potential errors to avoid:  confusing it
with other caves.  Not mixing up details from Lascaux or Chauvet.
Who discovered it and when?  I should check the date--early 20th century?  I think it was discovered in the
1920s, 1922 maybe, by teenagers or something similar.  Yes, it was discovered by three local teenagers, but
I'm not sure about their names.  Was it André David, Henri Dutertre, and Henri Breuil?  Or was there a Marthe
David?  Wait wait, I should be precise but not overly specific if unsure.  I'll not mention the names in the
final answer.

Now, let me confirm the prehistoric art discovered in the cave.  There were some of the oldest known human
stenciled handprints.  Some ancient footprints were also found.  They might have been from 9000 years ago, but
I'm not sure.  So I'll just mention the footprints without the date.  Oh right, there were some cave paintings
discovered as well, including images of animals such as horses, bison, mammoths, and deer.  Wait, was there
a bison?  Let me not mention that and only mention the ones that I'm confident about, such as the "spotted
horses" panel.
I should also mention the conservation of the cave and visitation information.  It was first opened to the
public in 1926, yes, that's right.  Today, you can have guided tours I believe.  How long does the tour take?
Maybe I don't need that level of detail since I'm not sure.
Also, any association with UNESCO: many caves are part of the "Decorated Caves of the Paleolithic" UNESCO
listing.  Pech Merle might be included in that, so mention it if that's the case.

Putting it all together:  concise paragraphs with these details.  Make sure terminology is correct--Upper
Paleolithic period, Gravettian culture.  If unsure about the culture, maybe just stick to the time frame.
Alright, ready to draft the answer.

</think>
<answer>
The Pech Merle Cave is a prehistoric cave located in the commune of Cabrerets, in the Lot department of
southwestern France.  The cave is famous for its Upper Paleolithic cave paintings, which date back to around
25,000 years ago during the Gravettian period.

Here are some specific details and examples about the Pech Merle Cave:
- **Discovery**:  The cave was discovered in 1922 by three local teenagers.  Since then, numerous excavations
and studies have been conducted to uncover the secrets of this ancient site.
- **Location**:  The cave is situated near the village of Cabrerets in the Lot department of southwestern
France.
- **Age of the paintings**:  The cave paintings at Pech Merle are estimated to be around 25,000 years old,
dating back to the Upper Paleolithic era.
- **Types of paintings**:  The cave contains a variety of paintings, including images of animals such as
horses, mammoths, and deer.  There are also human figures, geometric shapes, and abstract signs.
- **Notable paintings**:  One of the most famous paintings at Pech Merle is the "Spotted Horses" panel, which
features two horses with distinctive spots.  Another notable painting is the "Red Hand Stencil", which is a
handprint made by a prehistoric artist using red ochre.
- **Conservation and Visitation**:  The cave was opened to the public in 1926.  Today, visitors to the cave
can take a guided tour.  Visitors are limited to 700 per day to prevent overcrowding.
- **Archaeological significance**:  The Pech Merle Cave is considered one of the most important prehistoric
sites in France, providing valuable insights into the lives and culture of early humans in Europe.
- **UNESCO Recognition**:  The cave's unique and well-preserved artwork has earned it recognition as part of
the "Prehistoric Sites and Decorated Caves of the Vézère Valley" UNESCO World Heritage listing.

Overall, the Pech Merle Cave is an important cultural and historical site that offers a unique glimpse into
the lives of our prehistoric ancestors.
</answer>

Now answer the following question.  If it's a fact-seeking question, provide as many specific details and
examples as possible (such as names of people, numbers, events, locations, dates, times, etc.)

User:  <QUESTION_HERE>
```

*Figure 8.* (Cont.) LLM prompt for synthetic fact-seeking prompt generation.

Prompt for the LLM judge in the answer relevance reward ($\mathcal{R}_{rel}$)

```
You are given a user question and a response from an AI assistant. You have two
tasks. Your first task is to provide your own response to the user's question to
the best of your capability. Then, your second task is to act as an impartial judge
and evaluate whether your response or the given response from the AI assistant better
follows the user's instructions and provides a higher-quality answer.

You should mention your evaluation criteria for a high-quality response and a
detailed comparison of the two responses. Be explicit in the criteria you use and
explain how each response aligns with or deviates from them. Your judgement does not
need to focus on the factuality of the responses, and you should not try to verify
the correctness of any facts mentioned in the responses.

Avoid any biases towards either your own response or the given one. Do not allow the
length of the responses to influence your evaluation. Do not favor certain names of
the assistants. Be as objective as possible.

IMPORTANT: Provide your final verdict within <answer> and </answer> tags, strictly
following this format:
- <answer> [[A]] </answer> if the given response is better
- <answer> [[B]] </answer> if your own response is better

Below are the user's question and the response:

[User Question]
{instruction}

[The Start of the Assistant's Answer]
{response}
[The End of the Assistant's Answer]
```

*Figure 9.* LLM prompt for judging the answer quality in the answer relevance reward.

LLM Prompt for analyzing CoT traces

```
Below is a question and response generated by an LLM. Your task is to summarize
the reasoning process used by the LLM. Read the thought process carefully, and
annotate the explorations in the thought process used by the LLM. Specifically, write
down detailed steps the LLM took to pursue its thinking process, identifying all
meta-reasoning strategies used at each step, e.g. self-verification, backtracking,
summarization, etc.        Based on these analysis, also check the helpfulness of the
reasoning traces, e.g. in what ways and to what extent does the reasoning trace help
producing a more factual, detailed, and relevant answer. Derive a helpfulness_score
in the end.
The helpfulness_score should be derived on a scale of 0 to 10. Score 0 means the
thinking process is not useful at all. Score 10 means the reasoning traces are very
effective, leading to a significantly better answer.
Organize your answer in a json so that the steps and meta-reasoning strategies
(in field 'reasoning_strategies') and the final 'helpfulness_score' can be easily
extracted.

Question: {question}

Response from LLM: {response}
```

*Figure 10.* LLM prompt for analyzing the meta-reasoning strategies in the CoT traces.

