# OpenReview forum: "Learning to Reason for Factuality"
_ICML.cc/2026/Conference — ICML 2026 regular_

### Official Review · Reviewer_KqF2 · 2026-03-01

**Soundness:** 3
**Presentation:** 3
**Significance:** 4
**Originality:** 3
**Overall Recommendation:** 5
**Confidence:** 4

**Summary:**

This paper studies factuality in reasoning LLMs and argues that current reasoning-oriented RL training improves math/coding performance but can worsen long-form hallucination. The paper first shows that existing reasoning models such as QwQ-32B and DeepSeek-R1 underperform their non-reasoning counterparts on six long-form factuality benchmarks. It then proposes an online RL recipe for long-form factuality built on GRPO with a composite reward combining (i) factual precision from VeriScore-style claim verification, (ii) a detail-level term based on the number of supported claims, and (iii) an answer-relevance / general-quality term from pairwise LLM judging. The paper also introduces a scalable implementation of VeriScore to make online RL practical. Experiments on six benchmarks show large gains over the Llama-3.1-8B-Instruct base model in factual precision, improved detail level, and a modestly positive helpfulness win rate over the base model.

**Compliance With Llm Reviewing Policy:**

Affirmed.

**Final Justification:**

My final recommendation remains accept. I find the paper technically solid and practically relevant: it identifies an important failure mode of reasoning-oriented RL for long-form factuality, proposes a well-motivated composite reward to mitigate reward hacking, and validates the approach on a reasonably broad set of six benchmarks with useful ablations and offline baselines. In my view, the main contribution is not a fundamentally new RL algorithm, but rather a strong reward-design and systems contribution that makes online RL for long-form factuality practical and effective. This is a meaningful contribution for the community.

I agree with several concerns raised by other reviewers, particularly the heavy reliance on automatic evaluators and LLM judges, the absence of human evaluation, and the fact that the method is demonstrated on a single training backbone. These are real limitations and should be stated clearly. At the same time, I do not view them as fatal flaws in this setting. For long-form factuality, scalable and reliable evaluation is itself an open problem, and fully human evaluation is difficult to carry out at the scale needed for this kind of online RL study. Similarly, while additional backbones would certainly strengthen the paper, I do not think the current single-backbone evaluation is enough to negate the empirical contribution that is already present. Overall, these issues reduce the breadth of the claims more than they undermine the core technical results.

The rebuttal addressed my main concerns partially but meaningfully. In particular, the authors clarified the rationale for using a weaker training-time judge and a stronger evaluation-time judge, and clarified that the helpfulness result should be interpreted mainly as evidence that the method does not substantially degrade answer quality relative to the base model, rather than as a strong claim of superiority. These clarifications do not fully eliminate the dependence on automatic evaluation or the limited transfer evidence, but they make the paper’s claims more precise and the design choices more defensible. Overall, the rebuttal reinforced rather than changed my prior assessment: despite clear limitations, I still believe the paper’s problem framing, empirical contribution, and practical value are sufficient for acceptance.

**Key Questions For Authors:**

How robust are the main conclusions to the choice of evaluator / judge? In particular, can the authors report sensitivity to using different claim verifiers, different LLM judges, or different prompts for the relevance reward? A positive robustness result would increase my confidence in the soundness of the claims.

How much of the gain comes from online RL itself versus the specific reward design? The ablations are helpful, but it would be useful to isolate whether a stronger offline baseline or alternative online objective could recover similar gains. This would clarify whether the contribution is mainly “online RL matters” or “this specific reward matters.”

**Limitations:**

No. The paper does not appear to include an explicit limitations / broader-impact discussion. It would benefit from one that clearly acknowledges at least: (i) dependence on automatic evaluators and judges, (ii) absence of human evaluation, (iii) the possibility that optimizing to a particular verification stack could bias model behavior toward that stack’s preferences.

**Strengths And Weaknesses:**

Strengths

Soundness: The paper identifies a real technical issue in adapting online RL to long-form factuality: optimizing only factual precision encourages shorter answers, while optimizing factuality without relevance control can reward tangential yet correct content. The three-part reward is a sensible response to these failure modes, and the ablation in Table 2 is useful because it demonstrates that factuality-only rewards can indeed hurt answer quality while the relevance term recovers win rate. The empirical study is reasonably broad for this setting, spanning six long-form factuality benchmarks and including comparisons to existing reasoning and non-reasoning models, offline baselines, and reward ablations.

Presentation: The overall narrative is easy to follow: the paper motivates why long-form factuality is harder than standard RLVR tasks, explains why naive use of factuality metrics can be hacked, introduces the composite reward, and then validates it empirically. Figure 1, Table 1, and Table 2 are especially helpful for understanding the motivation and the precision/detail/relevance tradeoff. The paper also does a good job of making the reward definition concrete and operational.

Significance: Improving factuality in reasoning models is an important problem. If reasoning-oriented RL indeed increases hallucination in long-form responses, then methods that restore factual reliability without sacrificing utility are highly relevant to deployed systems. The proposed method makes the contribution potentially impactful for post-training practice.

Originality: The most original part of the paper is not any single ingredient in isolation, but the adaptation of online RL to the long-form factuality setting, where reward design is genuinely harder than in short-form QA or standard verifiable tasks. Relative to VeriScore, which is an evaluator rather than a training method, the paper contributes a scalable integration of such verification into the RL inner loop. The paper also remains meaningfully distinct from concurrent factuality-aware RL work such as [FSPO](https://arxiv.org/abs/2505.24630) and KnowRL, which emphasize step-wise factual supervision in reasoning traces rather than the paper’s long-form answer-level reward design.


Weaknesses

Soundness: My main concern is that the core conclusions depend heavily on automatic evaluators and judges. Both training and evaluation rely on claim extraction / verification pipelines related to VeriScore, and helpfulness is measured with a GPT-4o judge; moreover, the relevance reward during training is produced by prompting the base model to generate a reference answer and then judge against it.

Presentation: Writing is generally clear, but it would benefit from a careful proofreading pass. I noticed several small writing issues, for example: “ii) incentivize factuality” (subject-verb mismatch), “satisfing” (typo for “satisfying”), and “issue relevant search queries … and inspects the search results” (might be “inspect”). In addition, Figure 1 is hard to read: several text elements inside the boxes are too small and visually thin, which hurts legibility even though the figure is conceptually useful.

Originality: The paper is clearly novel at the level of problem setting (online RL for long-form factuality), the technical ingredients are individually less surprising: verifier-based factuality scoring, a detail-level shaping term, an LLM-judge relevance signal, and GRPO are all sensible extensions of existing ideas rather than a fundamentally new learning principle. In that sense, the novelty is best understood as a strong systems-and-reward-design contribution for a difficult setting, rather than a major algorithmic leap.

---

> ### Author Rebuttal · Authors · 2026-03-26
>
> Thank you for reviewing our paper. We appreciate your positive assessment of our work. Please find our detailed response below.
>
> ## W1: Soundness
> We acknowledge that our work relies heavily on automatic evaluators and LLM judges, which is indeed a limitation not only to this paper but to long-form factuality in general. Human evaluation on long-form factuality is extremely time-consuming and expensive – \\$4 per response according to Min et al. (2023) which amounts to \\$7k for evaluating a single run on the 6 benchmarks used in this paper. Nevertheless, we would like to provide additional insights and rationales for certain technical decisions made in this paper.
>
> > training and evaluation rely on VeriScore
>
> We agree that automated factuality evaluation metrics such as VeriScore are far from perfect, and accurately evaluating long-form factuality remains an open research problem. At present, we are not aware of a better alternative to VeriScore, and developing a new metric is out of scope for this work.
> In addition, we intentionally choose to use our scalable implementation only during training while sticking to the original VeriScore for evaluation. By showing substantial improvements in factuality using the standard, widely-used VeriScore in our final evaluation (while training on our own scalable VeriScore judge), it serves as partial evidence that we are not simply overfitting to a specific VeriScore judge.
>
> > helpfulness is measured with a GPT-4o judge
>
> We would like to ensure the quality of the final evaluation by using a SoTA frontier model (GPT-4o). In addition, the fact that a much weaker reward judge (Llama 3.1 8B) used during training still results in substantial performance improvements as evaluated by GPT-4o serves as evidence that our method is relatively stable and generalizable across different choices of LLM judges.
>
> > the relevance reward during training is produced by prompting the base model to generate a reference answer and then judge against it
>
> This allows us to use a simple yet effective binary reward to track whether our policy model maintains the same answer quality as the base model, which is shown to be very effective in our experiments (Table 2). While we acknowledge that there are alternative approaches for designing the answer relevance reward, we respectfully argue that our choice is principled, simple yet elegant.
>
> We will try to make these points clearer in the final version of our paper.
>
> ## W2: Presentation
> We thank the reviewer again for the careful review of our paper, and will make sure to fix these writing issues in the final version.
>
> ## Q1:
> Due to the cost of online RL training and automated factuality evaluation, we were unable to conduct systematic ablations on each of these points. As mentioned in the W1 discussion above, we tried our best to alleviate (but admittedly not fully resolve) some of these concerns.
>
> > different claim verifiers
>
> We adopt the claim extractor and claim verifier in the original VeriScore for our evaluation, and use Llama 3.3 70B in our scalable VeriScore during training, hoping to show that our method is not hacking against a certain VeriScore judge.
>
> > different LLM judges
>
> We use GPT-4o in our final evaluation and Llama 3.1 8B in training for a similar reason, serving as evidence that our method is relatively stable and generalizable across different choices of LLM judges.
>
> > different prompts for the relevance reward
>
> We acknowledge that there are alternative approaches for designing the answer relevance reward. Unfortunately we did not explore further in this direction since our relevance judge provides a very direct signal of whether our RL training degrades the helpfulness of the base model and proves to be effective in our experiments.
>
> ## Q2:
> This is a great question. We believe the answer is a combination of both. As we argue in the Introduction, we posit that online RL is critical for factuality as there has been a large body of research highlighting the importance of training on on-policy data for factuality purposes (see Section 6 for references). However, the journey of applying online RL to long-form factuality proved to be very challenging with severe reward hacking happening if we simply apply the reward used in standard offline RL (e.g. DPO). The journey of finding an effective and scalable reward formulation for online RL in long-form factuality constitutes the main contribution of this paper, which we hope to be useful for the research community as the first online RL recipe for long-form factuality. Conducting a more rigorous examination to separate the impact of online RL vs. the exact reward formulation would indeed be a great direction for future work.
>
> ## Limitations:
> We thank the reviewer for the insightful discussion and will add more discussions on the limitations of our approach to the final version of the paper.

---

> > ### Author Rebuttal · Reviewer_KqF2 · 2026-04-03
> >
> > I have read the authors’ rebuttal carefully. The response does not fully eliminate my concern about the heavy reliance on automatic evaluators and LLM judges, but it provides a reasonable clarification of the design choices and better explains why the reported gains are unlikely to be purely due to overfitting to a single evaluation stack. Overall, the rebuttal is consistent with my original assessment and does not materially change my score.

---

### Official Review · Reviewer_e8gM · 2026-03-12

**Soundness:** 3
**Presentation:** 3
**Significance:** 3
**Originality:** 2
**Overall Recommendation:** 4
**Confidence:** 3

**Summary:**

This paper applies online RL (GRPO) to improve long-form factuality in reasoning LLMs. The authors first show that `QwQ-32B` and `DeepSeek-R1` hallucinate more than their non-reasoning counterparts on six benchmarks. They then propose a composite reward combining factual precision (via a scalable VeriScore), detail level, and answer relevance (LLM-as-a-judge), and train `Llama-3.1-8B-Instruct` with this setup. Results: +23.1 pts average factual precision, 23% more supported claims, 54.4% GPT-4o win rate over the base model.

**Compliance With Llm Reviewing Policy:**

Affirmed.

**Key Questions For Authors:**

1. How closely does the scalable VeriScore agree with the original VeriScore? Any form of correlation or rank agreement would be helpful.

2. Have you tried a different judge for `Rrel`? The current setup where the base model judges its own outputs seems problematic.

3. Any variance or significance estimates for the win-rate results?

**Limitations:**

Some limitations need to be discussed:

1. Only tested on one 8B model. Transferability unknown.

2. All quality evaluation is LLM-judge-based. No human evaluation anywhere.

3. Reward pipeline depends on Google Search (Serper API) — not reproducible over time.

4. Models trained this way could learn to optimize specifically for VeriScore rather than genuine factuality.

The impact statement is quite minimal.

**Strengths And Weaknesses:**

Strengths:

1. **Well-motivated problem.** Long-form factuality for reasoning models is underexplored. The paper motivates this empirically — Table 1 shows both reasoning models hallucinate more than their non-reasoning counterparts across all six benchmarks.

2. **The reward ablation is the strongest part of the paper.** Table 2 is quite convincing: optimizing `Rfact` alone tanks relevance (Win Rate 36.9 with `Rdtl`), while `Rrel` largely fixes this. The rest of the contribution is more incremental.

3. **Broad evaluation.** Six benchmarks, comparison against same-base baselines (`SFT`, `DPO`, `GRPO`). The DPO baseline is particularly useful — it exposes a precision-vs-quality trade-off that offline RL misses.


Weaknesses:

The main concern is the evaluation methodology.

1. **Novelty is moderate.** The method combines GRPO, VeriScore, and an LLM judge. No new algorithm or verification method. The contribution is a training recipe and a faster VeriScore.

2. **I am not fully convinced by the helpfulness claim.** It relies entirely on a 54.4% GPT-4o win rate — no human eval, no variance estimates. Hard to tell if this is statistically meaningful. The `Rrel` reward also uses the base model as both reference generator and judge (Appendix G), which could bias the system toward base-model-like answers.

3. **The training-time reward and the evaluation metric are different implementations, and this is never validated.** Training uses `Llama-3.3-70B-Instruct` for claim extraction; evaluation uses the original VeriScore's finetuned models (Appendix B). It is unclear whether these produce consistent scores. This seems like an important gap.

4. **Single base model.** All experiments use `Llama-3.1-8B-Instruct`. It is unclear whether the recipe transfers.

Suggestions:

1. Report correlation or rank agreement between the training-time and evaluation-time VeriScore on a held-out set. This should be straightforward.

2. At minimum, add confidence intervals for the 54.4% win rate. A small human eval (50–100 examples) would also help.

3. Even partial results on a second base model (e.g. Qwen or Mistral) would help demonstrate that the recipe generallizes.

---

> ### Author Rebuttal · Authors · 2026-03-26
>
> Thank you for reviewing our paper! Please find our detailed response below.
>
> ## W1:
> While novelty assessment is subjective and we agree with the reviewer that the contribution of our paper is not new training algorithms, we would like to add that the novelty of this work lies in:
> 1) We show that existing SoTA R-LLMs suffer from significantly more hallucinations than their non-reasoning counterparts, highlighting an important open problem that has been overlooked hitherto.
> 2) We further propose the first Online RL recipe for long-form factuality with a novel reward formulation, and show that it achieves significant improvements over the base model as well as offline methods such as DPO (e.g. FLAME and FactTune).
>
> ## W2 & S2 & Q2:
> Our goal in this work is to improve the factuality of a given LLM in long-form generation through our proposed RL framework, **without degrading** the helpfulness, relevance, or general quality of that LLM (i.e. the base model). As a result, there are a few points we would like to clarify regarding your question:
> 1) Our goal is not to **beat** the base model in terms of helpfulness, but rather to maintain the same level (while greatly improving factuality). It is not our goal or intention to claim the 54.4% win rate indicating our model has better helpfulness than the base model with statistical significance. Instead, we argue it is strong evidence that our model is **not worse** than the base model.
> 2) Given that our goal is to prevent degradation in helpfulness against the base model, using the base model itself to produce the reference response allows us to use a simple yet effective binary reward to track whether our policy model maintains the same answer quality as the base model, which works well in our experiments (Table 2). We concur that using the base model to serve as both the reference generator and the judge could lead to potential biases. However, the judge model is more likely to favor its own responses in such cases, which only makes the judge stricter than an unbiased judge – we believe this is not detrimental to our goal.
>
> ## W3 & S1 & L2:
> We appreciate the reviewer for pointing out this issue. We choose to use our scalable implementation only during training while sticking to the original VeriScore for evaluation so that our final evaluation stays consistent with the literature (see our response W1 to dLzw). The only alternative would be to use our scalable implementation for both training and evaluation, but we dismissed it because it would i) make our evaluation inconsistent with existing papers, and ii) create a serious question whether we are overfitting to the particular VeriScore judge we are using. By showing substantial improvements in factuality using the standard, widely-used VeriScore in our final evaluation (while training on our own scalable VeriScore judge), it vindicates both the validity of our scalable VeriScore implementation and the effectiveness of approach.
>
> ## W4 & S3 & L1:
> Please see our response W2 to Reviewer dLzw.
>
> ## Q1:
> We only conducted limited experiments to validate the accuracy of our Scalable VeriScore in early stages of this project. In particular, we ran both the original and our scalable VeriScore for Llama 3.3 70B responses on the validation split of LongFact, and got the following results.
> - Original VeriScore Precision: 80.2%
> - Scalable VeriScore Precision: 75.4%
>
> Based on our examination of the results, we conclude that our Scalable VeriScore judge tends to be more accurate, because of the stronger LLM judge it uses. (The original VeriScore uses LoRA finetuned Mistral 7B claim extractor and Llama 8B claim verifier.) However, we stick to the original VeriScore for our final evaluation for reasons stated above.
>
> ## L3:
> The reviewer is absolutely correct that the dependence on Google Search makes the results less reproducible over time. However, given the nature of the factuality problem in open-ended long-form generations, we are not aware of any better alternative at this time. The main alternative is to rely on a fixed document index as the source of evidence, but that has been shown to be significantly inferior to the quality and comprehensiveness of the best commercial search engine (i.e. Google Search).
>
> ## L4:
> This is again a valid concern. As discussed in W3 above, we argue that this issue is partially mitigated—though certainly not resolved—by using different judges for VeriScore during training and evaluation. More broadly, accurately evaluating long-form factuality remains an open research problem and warrants dedicated investigation. At present, we are not aware of a better alternative to VeriScore, and developing a new metric is out of scope for this work.
>
> We will add a more detailed discussion on the limitations of our work to the final paper.

---

> > ### Author Rebuttal · Reviewer_e8gM · 2026-04-03
> >
> > Further experiments regarding suggestions 2 and 3 will help. Currently I will maintain my score.

---

### Official Review · Reviewer_6gG4 · 2026-03-13

**Soundness:** 3
**Presentation:** 2
**Significance:** 3
**Originality:** 2
**Overall Recommendation:** 3
**Confidence:** 2

**Summary:**

This paper addresses a critical gap in Reasoning Large Language Models (R-LLMs) research: the trade-off between enhanced reasoning capabilities and factual accuracy. The paper proposes Factual Reinforcement Learning from Automated Fact-Checking Feedback , a novel training approach that combines online RL with automated fact-checking feedback. Experiments show that this method substantially improves factuality with computational efficiency superior to baseline online RLHF methods.

**Compliance With Llm Reviewing Policy:**

Affirmed.

**Final Justification:**

maintain the original score

**Key Questions For Authors:**

See above.

**Limitations:**

See above.

**Strengths And Weaknesses:**

**Strengths:**

1. The paper tackles an important and timely issue regarding factuality in LLM reasoning processes, effectively expanding the application boundaries of reinforcement learning in reasoning tasks.
2. The experimental design is thorough and well-executed, with detailed analyses that effectively demonstrate the efficacy of the proposed method.
3. The paper is well-structured with clear writing and logical flow, making the content accessible and easy to follow.

**Weaknesses:**

1. The proposed approach lacks sufficient novelty, as similar ideas have been explored in previous work (i.e. VeriFastScore) .
2. While reward design is central to the proposed method, the paper fails to provide formal theoretical justification or mathematical proof for the reward mechanism. A rigorous formalization would strengthen the theoretical contribution and provide deeper insights into why this particular reward structure is effective.
3. The reward mechanism heavily depends on the quality of responses generated by the reference model, yet the paper does not provide detailed analysis of this critical dependency.

---

> ### Author Rebuttal · Authors · 2026-03-25
>
> Thank you for reviewing our paper! Please find our detailed response below.
>
> ## W1:
> While novelty assessment is subjective, we respectfully argue that our paper’s contribution differs significantly from VeriFastScore. VeriFastScore focuses solely on improving the speed of VeriScore, which is technically a “byproduct” of our paper. Our main contribution is proposing **the first** Online RL framework for long-form factuality with a novel multi-component reward design that effectively tackles reward hacking. To make VeriScore practical in online RL, we developed a faster and more scalable version from an engineering perspective, which is both faster than VeriFastScore (30x vs. 6.6x) and complementary to it. Further efficiency improvement may be achieved by leveraging the combined claim extraction and verification model from VeriFastScore in our scalable implementation, but that is out of scope for this paper.
>
> ## W2:
> We agree that the reward design is central and merits deeper justification, but we respectfully clarify that mathematical proofs for long-form factuality rewards are currently intractable. Unlike domains such as mathematics or coding, where answers can be deterministically verified through symbolic computation or code execution, there is no precise mathematical method to verify the factual accuracy of open-ended text. Because factuality in this context must rely on complex automatic evaluation frameworks like VeriScore, our contribution is fundamentally empirical.
>
> In the absence of theoretical guarantees, we provide “rigorous formalization” through formal definitions and extensive ablation studies: Section 3 formally defines the online RL objective, the full reward function (including each component), and the GRPO optimization loss; Section 3.1 provides the design rationale for each term; and Table 2 presents component-wise ablations. These ablations provide the requested deep insights by proving that all three reward components are practically necessary to align the model effectively in a complex, non-deterministic environment.
>
> ## W3:
> We thank the reviewer for raising this point. In the answer relevance reward, we adopt the response from the base model as the reference, for the following reasons. Our goal in this work is to improve the factuality of a given LLM in long-form generation through our proposed RL framework, without degrading the helpfulness, relevance, or general quality of that LLM (i.e. the base model). As a result, adopting the base model itself as the reference allows us to use a simple yet effective binary reward to track whether our policy model maintains the same answer quality as the base model, which is shown to be very effective in our experiments (Table 2). While we acknowledge that there are alternative approaches for designing the answer relevance reward, we argue that our choice is principled, simple yet elegant.

---

> > ### Author Rebuttal · Reviewer_6gG4 · 2026-04-04
> >
> > Thank you to the authors for the comprehensive response, but I still believe that there is a lack of effective comparative experiments and theoretical analysis, so I will maintain my score.

---

### Official Review · Reviewer_dLzw · 2026-03-16

**Soundness:** 3
**Presentation:** 2
**Significance:** 2
**Originality:** 2
**Overall Recommendation:** 3
**Confidence:** 4

**Summary:**

This paper studies factuality in reasoning LLMs and asks whether online RL can learn reasoning strategies that improve long form factual accuracy. The core idea is a new reward that combines factual precision, answer detail, and answer relevance, together with a scalable VeriScore implementation for online training. The paper trains a factual reasoning model from Llama 3.1 8B Instruct using SFT and GRPO, and evaluates on six long form factuality benchmarks. The main reported result is that the GRPO model improves average factual precision from 45.0 to 68.1, increases supported claims from 23.5 to 29.0, and reaches a 54.4 percent win rate against the base model on AlpacaEval style helpfulness judgments.

**Compliance With Llm Reviewing Policy:**

Affirmed.

**Key Questions For Authors:**

1. How sensitive are the main results to the choice of judge in the relevance reward and in the final helpfulness evaluation. Have the authors tried a stronger external judge for the reward term rather than the base model itself.

2. How much of the improvement comes from the reward design versus the synthetic prompt generation pipeline. Could the authors run a control with a different prompt source or a smaller synthetic set.

**Strengths And Weaknesses:**

Strength:
1. The paper addresses a genuinely important problem. It highlights that existing reasoning models can be worse than non reasoning counterparts on long form factuality, which is a useful and practically relevant observation.

2. The reward design is well motivated. The discussion of why factual precision alone is vulnerable to shorter answers and why detail alone can encourage tangential content is clear and compelling.

3. The experimental scope is solid. Evaluation covers six long form factuality benchmarks, includes comparisons to existing reasoning models and offline baselines, and reports both factuality and helpfulness related metrics.

Weakness:

1. The helpfulness and relevance evaluation relies heavily on LLM judging. The paper uses GPT 4o for win rate evaluation, and the reward itself uses the base model as a judge for answer relevance. This is convenient, but it raises concerns about bias and calibration, especially because the appendix explicitly notes that this design limits the choice of judge and that stronger judges were not explored.

2. The method is demonstrated on only one training backbone, namely Llama 3.1 8B Instruct. Given that the paper motivates the work partly through failures of larger reasoning models such as DeepSeek R1 and QwQ, it would be stronger to show at least one additional training backbone or a more direct transfer result.

3. The paper depends on synthetic training prompts generated with Llama 4, and the robustness of the approach to different prompt distributions is not deeply tested. That makes it harder to know how much of the gain comes from the reward design versus the prompt curation pipeline.

4. Some reward hacking remains. In the ablations, increasing the detail reward can still push the model into worse relevance tradeoffs, with win rate dropping to 45.7 percent. This does not break the paper, but it shows the method is not yet fully stable.

---

> ### Author Rebuttal · Authors · 2026-03-25
>
> Thank you for reviewing our paper! Please find our detailed response below.
>
> ## W1 & Q1:
> We acknowledge that our evaluation relies heavily on LLM judging, as human evaluation on long-form factuality is extremely time-consuming and expensive – \\$4 per response according to Min et al. (2023) which amounts to \\$7k for evaluating a single run on the 6 benchmarks used in this paper. We further acknowledge that more experiments would be helpful to systematically study the impact of various reward judges. Nonetheless, we would like to highlight some considerations regarding our choice of “using GPT-4o for evaluation and Llama 3.1 8B as the reward for training”. We intentionally choose a much stronger judge for evaluation and a weaker one for training for a few reasons: First, we would like to ensure the quality of the final evaluation by using a SoTA frontier model (GPT-4o). In addition, the fact that a much weaker reward judge (Llama 3.1 8B) used during training still results in substantial performance improvements as evaluated by GPT-4o serves as evidence that our method is relatively stable and generalizable across different choices of LLM judges.
>
> ## W2:
> Given the high cost of online RL training, we opt to conduct comprehensive experiments and ablations using a single backbone architecture, which, to our delight, is unanimously praised by all reviewers. On the other hand, there has been a follow-up paper from the research community which cited (the preprint version of) our paper and applied (a simplified variant of) our approach to Qwen 3 4B and 8B. It reduces the hallucination rate of the Qwen 3 baseline by 24.5 points on Biography, 18.1 points on WildHallu, 27.6 points on PopQA, and 8.9 points on GPQA. This serves as direct and strong evidence that our approach transfers well to other backbone architectures. (We will not provide a link to that paper as it is not clear to us whether that would be a violation of the anonymity requirement, because that paper cites the unanonymized version of our paper.)
>
> ## W3 & Q2:
> We thank the reviewer for raising another great point. As discussed in Section 2.1, we find that the previous approach of filtering existing instruction tuning datasets to select fact-seeking questions (Lin et al., 2024) resulted in many noisy and low-quality prompts. As a result, we propose the synthetic prompt generation method grounded on two distinct sets of natural prompts. Unfortunately, we were unable to experiment with multiple prompt curation approaches because of the prohibitive cost of the entire training pipeline. Our intuition is, however, that if our method performs well on real-world benchmarks by training only on synthetic prompts, it would hopefully achieve similar or better results if someone manages to generate a higher-quality synthetic prompt set or curate a high-quality fact-seeking dataset from natural data.
>
> ## W4:
> While it is certainly possible that reward hacking still happens to some extent with our model, we respectfully argue that “the win rate drops to 45.7% when increasing the detail reward (by 10x)” is expected and does not undermine the validity of our method. In particular, when assigning a higher weight to the detail reward, the model is incentivized to produce longer and sometimes less pertinent or more repetitive responses, which contains more of both supported and unsupported claims. Furthermore, the supported claims would become increasingly tangential to the question, as discussed in detail in the paper. As a result, the general answer quality degrades and is reflected in the lower win rate. In our opinion, this result is aligned with our explanation of reward hacking and indicates that our approach behaves as expected.

---

### Decision · Program_Chairs · 2026-04-30

**Decision:**

Accept (regular)

**Comment:**

This paper proposes a reward model that simultaneously improves the reasoning ability and factual correctness of reasoning LLMs.

Strength:
1. **Really important problem** (Reviewers dLzw, 6gG4, e8gM, KqF2)
2. Experimental scope is solid (Reviewers dLzw, 6gG4, e8gM)
3. Well-structured with clear writing (Reviewers 6gG4, KqF2)
4. Well-motivated design (Reviewer dLzw)

Weakness:
1. **Relies heavily on LLM judging** (Reviewers dLzw, KqF2) - This is not fully resolved during the rebuttal. While I understand that human evaluation could be expensive, it would still be possible to sample a smaller subset and show some human evaluation results.
2. **Only one training backbone** (Reviewers dLzw, e8gM) - It is not clear whether this is fully resolved. While the authors claim that others have implemented their method, it is difficult to check the paper without violating the anonymity requirement. It would be better if the authors had provided some additional experimental results conducted by themselves.
3. Robustness of different prompts (Reviewers dLzw, 6gG4)
4. Technical ingredients not surprising (Reviewers 6gG4, e8gM, KqF2)

I carefully read all reviews and the author responses, and find the responses **only partially resolve the reviewers' concerns**, as agreed by the reviewer who gave the highest score (5).  While those issues will not directly result in a rejection of the paper, I do believe addressing these problems will significantly improve the quality of the paper.